# LAG3-PD1 or CTLA4-PD1 Inhibition in Advanced Melanoma: Indirect Cross Comparisons of the CheckMate-067 and RELATIVITY-047 Trials

**DOI:** 10.3390/cancers14204975

**Published:** 2022-10-11

**Authors:** Bai-Wei Zhao, Fei-Yang Zhang, Yun Wang, Guo-Ming Chen, Man Nie, Zhou-Kai Zhao, Xiao-Jiang Chen, Kai-Ming Jiang, Run-Cong Nie, Ying-Bo Chen

**Affiliations:** 1Department of Gastric Surgery & Melanoma Surgical Section, State Key Laboratory of Oncology in South China, Collaborative Innovation Center for Cancer Medicine, Sun Yat-sen University Cancer Center, Guangzhou 510060, China; 2Department of Hematologic Oncology, State Key Laboratory of Oncology in South China, Collaborative Innovation Center for Cancer Medicine, Sun Yat-sen University Cancer Center, Guangzhou 510060, China; 3Department of Medical Oncology, State Key Laboratory of Oncology in South China, Collaborative Innovation Center for Cancer Medicine, Sun Yat-sen University Cancer Center, Guangzhou 510060, China

**Keywords:** PD-1, CTLA4, LAG3, melanoma, immunotherapy

## Abstract

**Simple Summary:**

In the past few decades, targeted therapy and immunotherapy have transformed tremendously the chances of survival. Checkpoint inhibitors that block programmed cell death 1 (PD-1) and cytotoxic T-lymphocyte antigen 4 (CTLA-4) pathways hold the most promise. However, more than half of the patients treated with CTLA-4-PD1 inhibition suffered from grade 3 or 4 treatment-related adverse events (TRAEs). Recently, Tawbi et al. reported the initial results of the phase 2–3 RELATIVITY-047 trial which evaluated LAG3-PD1 inhibition of relatlimab plus nivolumab in patients with previously untreated advanced melanoma. Here, we performed an indirect cross-comparison of LAG3-PD1 and CTLA4-PD1 inhibition in patients with previously untreated advanced melanoma by deriving individual patient survival data and safety profiles. We found that the PFS of LAG3-PD1 and CTLA4-PD1 inhibition were similar. Compared with CTLA4-PD1-inhibition, LAG3-PD1 inhibition tended to exhibit earlier survival benefit and lesser TRAEs.

**Abstract:**

**Objective:** To compare the inhibition of LAG3-PD1 versus the inhibition of CTLA-4-PD1 in patients with previously untreated advanced melanoma. **Methods:** The individual participant data (IPD) were extracted from the KM plots using a graphical reconstructive algorithm. Log-rank, Cox proportional hazard model, Bayesian hierarchical model with time-varying hazard ratio (HR) effect, and restricted mean survival time (RMST) were performed to estimate survival benefits. **Results:** The CheckMate-067 (*n* = 630) and RELATIVITY-047 (*n* = 714) trials were included for analysis. The graphical reconstructive algorithm showed that IPD had similar HRs and log-rank values as the original plots. The HR of nivolumab plus relatlimab (LAG3 inhibitor) versus nivolumab plus ipilimumab (CTLA4 inhibitor) was 1.19 (95% confidence interval [CI] 0.96 to1.48). The 24-months RMST of nivolumab plus relatlimab versus nivolumab was 2.35 (95% CI 0.77–3.94) months, compared with 1.87 (95% CI, 0.25–3.49) months for nivolumab plus ipilimumab versus nivolumab. The Bayesian hierarchical model showed that patients treated with nivolumab plus relatlimab had earlier PFS benefits than those with nivolumab plus ipilimumab. Grade 3 or 4 treatment-related adverse events occurred in 18.9% of patients using nivolumab plus relatlimab and 55.0% of patients using nivolumab plus ipilimumab. **Conclusions:** These findings suggest that the PFS of LAG3-PD1 and CTLA4-PD1 inhibition were similar and LAG3-PD1 inhibition exhibited earlier survival benefit and lesser TRAEs.

## 1. Introduction

In 2004, there were no systemic therapies associated with significant survival benefits in advanced melanoma [1]. However, in the past few decades, targeted therapy and immunotherapy have transformed tremendously the chances of patient’s survival [2,3,4,5]. Checkpoint inhibitors that block programmed cell death 1 (PD-1) and cytotoxic T-lymphocyte antigen 4 (CTLA-4) pathways hold the most promise [6]. Evidence from two randomized trials established dual CTLA4-PD1 inhibitors (nivolumab–ipilimumab) as the standard of care for previously untreated advanced melanoma [2,7]. Recently, the long-term outcome results of the CheckMate-067 trial showed that the median overall survival (OS) of patients treated with nivolumab plus ipilimumab could reach up to 72.1 months [2]. However, more than half of the patients treated with nivolumab plus ipilimumab suffered from grade 3 or 4 treatment-related adverse events (TRAEs) [7,8]. Therefore, there remain unmet needs to identify novel immune checkpoint inhibitors to improve the benefit–risk profile of immunotherapy.

The lymphocyte-activation gene 3 (LAG3) is a cell surface protein found on immune cells, especially on activated CD4+ and CD8+ T cells [9]. LAG3 contains specific domains that act as a high-affinity binding site for major histocompatibility complex (MHC) class II, functioning as an immune checkpoint, contributing to immune escape in carcinogenesis [10]. Although the underlying functional mechanism is yet to be clarified, LAG3 monoclonal antibodies are now being explored in cancer treatment. Recently, Tawbi et al. reported the initial results of the phase 2–3 RELATIVITY-047 trial which evaluated LAG3-PD1 inhibition of relatlimab plus nivolumab in patients with previously untreated advanced melanoma [11]. The findings suggested that, compared with PD1 inhibition alone, dual LAG3-PD1 inhibition could reduce the risk of disease progression or death by 25%. The median progression-free survival (PFS) was 10.1 months with LAG3-PD1 inhibition as compared with 4.6 months with PD1 inhibition alone (hazard ratio for progression or death, 0.75 [95% CI, 0.62–0.92]). The PFS of nivolumab plus relatlimab observed in the RELATIVITY-047 trial was similar to those of nivolumab plus ipilimumab in the CheckMate-067 trial. In addition, grade 3 or 4 TRAEs were reported in 18.9% of patients receiving nivolumab plus relatlimab, as compared with 55.0% in patients who received nivolumab plus ipilimumab. However, the comparison between LAG3-PD1 inhibition and CTLA4-PD1 inhibition was lacking.

In this study, we performed an indirect cross-comparison of LAG3-PD1 and CTLA4-PD1 inhibitions in patients with previously untreated advanced melanoma by deriving individual patient survival data and safety profiles.

## 2. Methods

### 2.1. Data Sources and Selection

Although this study is not a traditional meta-analysis directly comparing LAG3-PD1 with CTLA4-PD1 combination immunotherapy, it was conducted following the Preferred Reporting Items for a Systematic Review and Meta-analysis of Individual Participant Data (PRISMA-IPD) protocol [12].

A literature search was conducted on PubMed, Web of Science and Embase for eligible publications between 1 January 2015 and 14 January 2022 (R.-C.N. and Y.W.). We included phase 2 or 3 clinical trials that reported the use of dual LAG3-PD1 inhibitors or dual CTLA4-PD1 inhibitors, and assigned treatments using PD1 inhibitors alone as the control arms for first-line treatment of advanced or metastatic melanoma. Only trials that reported the Kaplan–Meier (KM) curves of treatments were included. Reviews, conference abstracts and non-English-language articles were excluded. To eliminate duplicated data, only the latest data from an eligible study were used.

### 2.2. Data Extraction and Synthesis

The following characteristics of the eligible trials were extracted: main eligibility criteria, intervention group, treatment dosing, survival outcomes, percentage of *BRAF* V600 mutation status, PD-L1 expression, and TRAEs. The data extraction was performed by two independent investigators (B.-W.Z. and F.-Y.Z.). The individual patient data were extracted and decoded from the reported KM curves using algorithms described by Guyot et al. [13]. The median PFS and hazard ratio (HR) of extracted IPD were compared with the original results of included publication. To improve the comparability of the included trials, survival events after 24 months were censored at 24 months.

### 2.3. Statistical Analysis

The primary endpoint was PFS, defined from the time of randomization until the earliest date of disease progression or death, whichever occurred first. In this study, three different approaches were used to quantify the survival benefits of dual LAG3-PD1 or CTLA4-PD1 combination: (1) Primary analysis was performed using the Bayesian hierarchical model with a time-varying HR effect [14]. In the Bayesian analysis, the time-varying HR effect was modeled by assuming that the hazards were constant within each 1-month follow-up and truncated results at 24 months. Therefore, each 1-month segment had its own hazard rate and HR. The posterior mean of PFS distributions was also calculated, using the Markov chain Monte Carlo methods [15]; (2) The log-rank test and marginal Cox model, derived from proportional hazards, were used to evaluate survival benefits, and; (3) Restricted mean survival time (RMST) was also used to estimate the average progression time free from an event up until a specific timepoint [16]. All statistical analyses were performed using the R software, version 4.2.0 (http://www.r-project.org, created by Ross Ihaka and Robert Gentleman from The University of Auckland) with packages IPD from KM, RcppHungarian, survRM2, survival, and rstanarm. In this study, *p* < 0.05 was considered for statistical significance.

## 3. Results

After the initial search of PubMed, Web of Science, and Embase (Figure 1), a total of 621 reports were identified. Removing the duplicated reports and initially screening the titles and abstracts, a total of 12 articles were eligible for full texts screened. Finally, only CheckMate-067 and RELATIVITY-047 trials were found to be eligible and therefore included for further analyses [2,11]. Both trials were conducted in previously untreated metastatic melanoma. CheckMate-067 was a phase 3 randomized trial that compared nivolumab plus ipilimumab or nivolumab alone with ipilimumab alone. RELATIVITY-047 was a phase 2–3 randomized trial that combined LAG3 and PD1 inhibition with nivolumab plus relatlimab compared with nivolumab alone. The characteristics of the two trials are presented in Table 1. The main inclusion and exclusion criteria were similar. RELATIVITY-047 contained patients with a higher percentage of the *BRAF* mutation and tumor PD-L1 positive status.

In order to validate the power of the reconstructive algorithm, the algorithm-yielded IPD and the original plots were compared. The graphical reconstructive algorithm-yielded IPD has similar HRs and log-rank values to the original plots. For CheckMate-067, the median PFS was 12.07 months and 6.91 months for nivolumab plus ipilimumab and nivolumab alone in the reconstructed IPD, versus 11.50 months and 6.90 months in the original report. The HRs were 0.78 (95% confidence interval [CI], 0.63–0.96) and 0.79 (95% CI, 0.65–0.97) for reconstructed and original PFS, respectively. For RELATIVITY-047, similar results were observed, indicating that the reconstructed IPD could effectively represent the original data.

Plotting the KM curves of the two trials together (Figure 2A), we found that the PFS of nivolumab plus relatlimab in the RELATIVITY-047 trial was slightly shorter than that of nivolumab plus ipilimumab in the CheckMate-067, with an HR of 1.19 (95% CI, 0.96–1.48; *p* = 0.118). The PFS of nivolumab was also found to be slightly shorter in the RELATIVITY-047, as compared with the CheckMate-067 (HR, 1.16 [95% CI, 0.95–1.41]; *p* = 0.157). The differences in 12- and 24-months RMST of nivolumab plus relatlimab versus nivolumab alone in the RELATIVITY-047 were 1.08 (95% CI, 0.37–1.79; *p* = 0.003) months and 2.35 (95% CI, 0.77–3.94; *p* = 0.004) months, compared with 1.19 (95% CI, 0.47–1.92; *p* = 0.001) months and 1.87 (95% CI, 0.25–3.49; *p* = 0.024) months for nivolumab plus ipilimumab versus nivolumab alone in the CheckMate-067 trial (Figure 2B).

A Bayesian hierarchical model was constructed to evaluate the time-varying survival benefits. Interestingly, nivolumab plus relatlimab in the RELATIVITY-047 trial was associated with a relatively early PFS benefit (7.40 vs. 11.50 months; Figure 3A). At 12 months, the estimated PFS was 46.4% (41.0–52.2%) and 39.6% (34.1–44.4%) for nivolumab plus relatlimab and nivolumab alone in the RELATIVITY-047 trial. Similarly, the estimated PFS was 51.5% (45.3–57.5%) and 50.0% (44.5–55.6%) for nivolumab plus ipilimumab and nivolumab alone in the CheckMate-067 trial (Figure 3B). At 24 months, the estimated PFS was 37.1% (30.9–43.6%) and 17.3% (12.4–21.8%) for nivolumab plus relatlimab and nivolumab alone in the RELATIVITY-047 trial. The estimated PFS was 42.1% (36.2–48.5%) and 30.2% (24.9–36.2%) for nivolumab plus ipilimumab and nivolumab alone in the CheckMate-067 trial (Figure 3B).

TRAEs of grade 3 or 4 occurred in 18.9% of patients in the nivolumab plus relatlimab group of the RELATIVITY-047 trial and in 55.0% of patients in the nivolumab plus ipilimumab of the CheckMate-067 trial. TRAEs leading to treatment discontinuation were lower in patients receiving nivolumab plus relatlimab than patients receiving nivolumab plus ipilimumab (14.6% vs. 36.4%) (Table 1).

## 4. Discussion

To the best of our knowledge, this is the first study to compare the dual LAG3-PD1 and CTLA4-PD1 inhibitions for advanced melanoma. Our findings showed that the PFS of dual LAG3-PD1 and CTLA4-PD1 inhibitions were similar. In addition, LAG3-PD1 inhibition tended to exhibit earlier survival benefit and have fewer TRAEs.

In the prior immunotherapy era, the prognosis of advanced melanoma was extremely dismal, with the median overall survival time less than 12 months due to poor therapeutic responses to the conventional chemotherapy or radiotherapy treatments [17,18]. In the past few decades, immune checkpoint inhibitors that target the dysfunctional immune microenvironment have revolutionized the treatment landscape of advanced melanomas, changing it from an incurable malignancy into a potentially curative disease [19]. PD1 inhibitors alone or their combination are now the standard systemic therapy care for advanced melanoma. Dual CTLA4-PD1 blockade with nivolumab plus ipilimumab have been approved by the US Food and Drug Administration (FDA) as first-line therapy based on the promising response and survival outcomes reported by the CheckMate-067 trial [2,8,20]. The long-term outcomes of the CheckMate-067 trial showed that the median OS of nivolumab plus ipilimumab could reach up to 72.1 months and the 6.5-year OS rates was 57% in patients with BRAF-mutant tumors. However, given the substantially higher risk of toxicity (grade 3–4 or TRAEs: 55.0%) with nivolumab plus ipilimumab, there is much concern about other immune checkpoint inhibitors, such as anti-LAG3 [21]. The recent RELATIVITY-047 phase 2–3 study reported on the dual inhibition of LAG3 and PD1 using a fixed dose of nivolumab plus relatlimab, as compared with nivolumab alone [11]. The LAG3 and PD1 combination was associated with a 25% reduction in the risk of disease progression or death. However, a cross-trial comparison of these two dual inhibitions is lacking but urgently needed.

In this study, we reconstructed the IPD through a graphical–reconstructive algorithm and found that the HRs and log-rank values were similar between reconstructed IPD data and the initially reported results, suggesting that the feasibility of our results warrant deeper comparisons of eligible trials. The merged KM curves showed that the curve of the nivolumab plus relatlimab group fell below that of nivolumab plus ipilimumab. However, the survival difference was not statistically significant (HR = 1.19, *p* = 0.118). It should be noted that the curve of the nivolumab alone group of the RELATIVITY-047 trial was also below that of the CheckMate-067 trial (HR = 1.16, *p* = 0.157). These results indicate the potential differences between these two clinical trials. This is partly because of the different evaluation methods of the two trials (CheckMate-067: investigator’s radiology evaluation; RELATIVITY-047: blinded independent review), and the higher proportion of BRAF mutation in the RELATIVITY-047 trial. Therefore, survival benefit using the same control reference (nivolumab alone) rather than a direct comparison of nivolumab plus relatlimab and nivolumab plus ipilimumab would be more clinically relevant.

The HRs based on the Cox proportional hazards (PHs) model in our study suggested that the PFS benefit were similar in the nivolumab plus relatlimab and nivolumab plus ipilimumab groups. Nonetheless, when treated with immunotherapy, the hazards were often non-proportional during the treatment period, the HRs derived from PHs assumption were thus, to some extent, not suitable. RMST, the expected time spent event free for future patients followed for the specified time point, is an alternative option that may overcome the limitations of PHs assumption [22,23]. The difference in RMST for the PFS of these two trials was evaluated in this present study. Our findings showed that the RMST difference estimated in the RELATIVITY-047 trial was parallel to those of the CheckMate-067 trial. Further, the RMST difference in the RELATIVITY-047 trial seemed to increase more quickly than that in the CheckMate-067 trial. In this study, we also used the Bayesian hierarchical model to assess the time-varying survival benefit of the two trials. We found that nivolumab plus relatlimab in the RELATIVITY-047 trial was associated with an earlier PFS benefit and a greater PFS benefit as the follow-up duration extended. Overall, these results indicated that nivolumab plus relatlimab may have a higher potential to exhibit a survival benefit.

Besides survival benefit, the risk of toxicity should also be considered. Notably, the incidences of TRAEs of any grade, grade 3–4 or TRAEs leading to treatment discontinuation were less frequent among patients who received nivolumab plus relatlimab than among patients who received nivolumab plus ipilimumab. Nonetheless, the TRAEs of the nivolumab group were also less common in the RELATIVITY-047 trial. The enrollment period of participants in the CheckMate-067 trial (year: 2013–2014) was much earlier than that of the RELATIVITY-047 trial (year: 2018–2020). It is probable that the increasing use of immunotherapy has increased clinicians’ ability to manage TRAEs [24], and thus, could partly explain the decreasing TRAEs observed in the RELATIVITY-047 trial. However, clinical trials that enrolled patients recently showed that the grade 3 or 4 TRAEs in patients receiving nivolumab plus ipilimumab still ranged from 30.0% to 52.5% [25,26,27,28,29], higher than that observed in the nivolumab plus relatlimab group of the RELATIVITY-047 trial. Therefore, it is believed that the safety profile of nivolumab plus relatlimab appears more favorable than that of nivolumab plus ipilimumab.

## 5. Limitations

The findings of this study should be interpreted with caution considering the naive univariate nature of the survival models. The first and the most notable limitation is that the patient-level data of these two trials were retracted from the KM plots of the two trials. We could not assess other covariates to make the indirect comparisons between two monotherapies by adjusting other covariates. In addition, we could not adjust other covariates to compare the dual LAG3-PD1 inhibition with CTLA4-PD1 inhibition. Even if we consider the group contrasts between the treated and the control to assess the treatment effect for the combinations, the KM plots of the two trials indicated that other pertinent patient-level covariates should be adjusted to directly compare dual LAG3-PD1 inhibition with CTLA4-PD1 inhibition. In addition, the debatable topic of whether patients with low LAG3 expression would benefit more from nivolumab–ipilimumab, and whether patients with *BRAF*-wild-type would benefit more from nivolumab plus relatlimab remains unsolved. Lastly, the follow-up of the RELATIVITY-047 trial is still immature, and overall survival was not reported; thus, whether the long-term PFS and overall survival benefit of nivolumab plus relatlimab is similar to or better than that of nivolumab plus ipilimumab remains to be elucidated. However, from the perspective of commercial interests, it is well recognized that it is unlikely that there will be a head-to-head randomized clinical trial to compare the two combinations. Therefore, the present study is to some extent critical for helping in the exploration of immunotherapy for advanced melanoma.

## 6. Conclusions

To the best of our knowledge, this is the first study to compare the dual LAG3-PD1 and CTLA4-PD1 inhibitions for advanced melanoma. IPD data suggest that the PFS of LAG3-PD1 and CTLA4-PD1 inhibitions were similar. In addition, compared to CTLA3-PD1 inhibition, LAG3-PD1 inhibition seems to exhibit earlier survival benefits and a lesser extent of TRAEs.

## Figures and Tables

**Figure 1 cancers-14-04975-f001:**
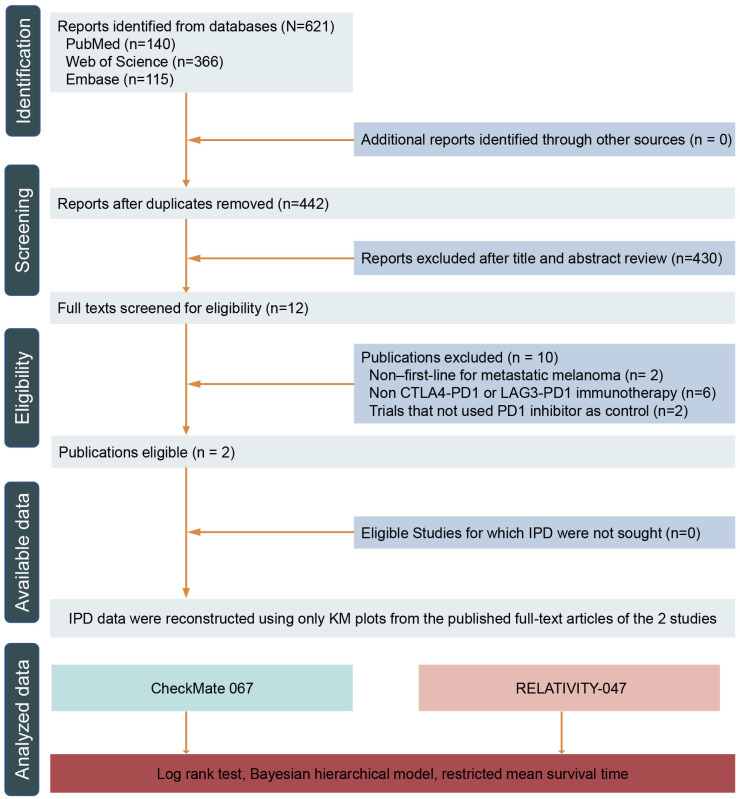
PRISMA flow chart for selecting eligible studies. Abbreviations: PD-1, programmed cell death 1; LAG3, Lymphocyte-activation gene 3; CTLA4, cytotoxic T-lymphocyte–associated antigen 4; IPD, individual patient data; KM, Kaplan–Meier.

**Figure 2 cancers-14-04975-f002:**
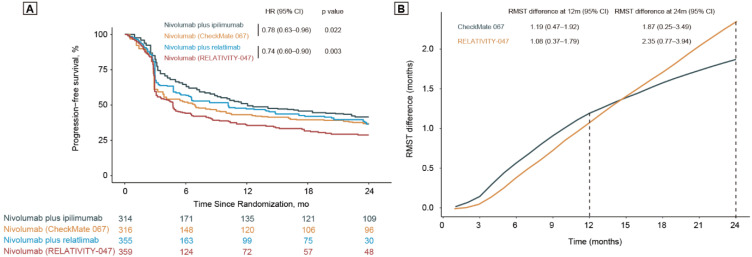
Survival analysis of the CheckMate-067 and RELATIVITY-047 trials. (**A**) Kaplan–Meier curves and log-rank test of CheckMate-067 and RELATIVITY-047 trials. The events that occurred after 24 months were censored to improve the comparability of these two trials. (**B**) The difference in RMST between experimental groups and control groups of CheckMate-067 and RELATIVITY-047 trials. Abbreviations: HR, hazard ratios; CI, confidence interval; RMST, restricted mean survival time.

**Figure 3 cancers-14-04975-f003:**
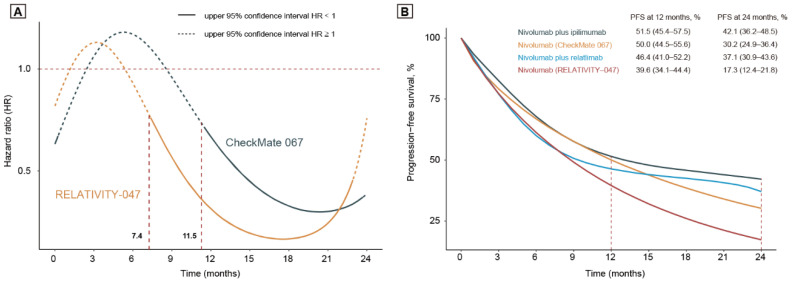
Bayesian Hierarchical Survival Model. (**A**) The time-varying HR is based on the Bayesian hierarchical model. The dotted line indicates that the upper limit of the confidence interval of estimated HR is higher than 1. (**B**) Posterior means of PFS of the treatment arms of CheckMate-067 and RELATIVITY-047 trials. Abbreviations: HR, hazard ratios; CI, confidence interval; RMST, restricted mean survival time.

**Table 1 cancers-14-04975-t001:** Study Information and Characteristic of Trials.

Trial, Year, Clinical Trials.Gov Identifier	No. of Countries	Main Eligibility Criteria	Intervention Groups, Dosing	PFS	BRAF Mutation	PD-L1 Expression	TRAEs	Total Included Cases
CheckMate-067 [2] Wolchok, 2022, NCT01844505	137 (Australia, Europe, Israel, New Zealand, and North America)	(1) Previously untreated unresectable stage III or IV (2) Age 18 years or older (3) ECOG PS 0-1 (4) Measurable disease as per RECIST 1.1 (5) Availability of tumor sample to assess PD-L1 status and BRAF V600 mutation	Nivolumab plus ipilimumab: nivolumab 1 mg/kg plus ipilimumab 3 mg/kg once every 3 weeks for 4 doses, followed by nivolumab 3 mg/kg once every 2 weeks for cycle 3 and beyond; Nivolumab: nivolumab 3 mg/kg once every 2 weeks (plus ipilimumab-matched placebo)	Median: 11.5 vs. 6.9 months; HR: 0.79 (0.66–0.97)	31.9% vs. 32.2%	23.5% vs. 21.7%	All TRAEs: 95.5% vs. 82.1%; TRAEs of Grade 3 or 4: 55.0% vs. 16.3% TRAEs leading to discontinuation: 36.4% vs. 7.7%	630
RELATIVITY-047 [11] Tawbi, 2022 NCT03470922	111 (North America, Central America, South America, Europe, Australia, and New Zealand)	(1) Previously untreated unresectable stage III or IV (2) Age 12 years or older (3) Measurable disease as per RECIST 1.1 (4) Availability of tumor sample to assess PD-L1 and LAG3 status	Relatlimab plus nivolumab: relatlimab 180 mg plus nivolumab 480 mg once every 4 weeks Nivolumab: nivolumab 480 mg once every 4 weeks	PFS: 10.12 vs. 4.63 months; HR: 0.75 (0.62–0.92)	38.3% vs. 38.7%	41.1% vs. 40.9%	All treatment-related adverse events: 81.1% vs. 69.9%; Treatment-related adverse events of Grade 3 or 4: 18.9% vs. 9.7%; TRAEs leading to discontinuation: 14.6% vs. 6.7%	714

Abbreviations: ECOG PS, Eastern Cooperative Oncology Group performance status; PD-L1, programmed death-ligand 1; LAG3, Lymphocyte-activation gene 3; PFS, progression-free survival; RECIST 1.1, Response Evaluation Criteria in Solid Tumors, version 1.1; TRAEs, treatment-related adverse events.

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
