# Peer review of "LAG3-PD1 or CTLA4-PD1 Inhibition in Advanced Melanoma: Indirect Cross Comparisons of the CheckMate-067 and RELATIVITY-047 Trials"

_cancers, 2022, doi:10.3390/cancers14204975_

Round 1

Reviewer 1 Report

This paper is very interesting and important for understanding the base of the recently described new modes of immunotherapy of melanoma.  at the same time the paper pretty well illustrates the applicability and usefulnest of numerical posterior analyses of the already published data.

I have only one remark, namely, please carefully scan the text and remove all the typos and grammar errors, as in parts of the text they are numerous.

Author Response

  1. This paper is very interesting and important for understanding the base of the recently described new modes of immunotherapy of melanoma.  at the same time the paper pretty well illustrates the applicability and usefulnest of numerical posterior analyses of the already published data.

Reply: Thanks very much for your comment.

I have only one remark, namely, please carefully scan the text and remove all the typos and grammar errors, as in parts of the text they are numerous.

Reply: Thanks very much for your comment. We had revised all the typos and grammar errors (Page 2, Line 32, 33, 35, 37-40; Page 3, Line 45, 51; Page 4, Line 71, 73; Page 4, Line 94, 122, 123; Page 5, Line 137; Page 7, Line 173; Page 9, Line 220; Page 10, Line 223, 239; Page 12, Line 305).

Reviewer 2 Report

The authors of this paper analysed two randomised trials with dual checkpoint inhibition in patients with metastatic melanoma to compare the efficacy of PD1-LAG3 vs PD1-CTLA4.

The comparison is based on the extraction of IPD from KM curves of the original studies, through a dedicated algorithm. The study is presented clearly, the analysis is precise, and the discussion is comprehensive.

My main concern relates to the fact that the same conclusions can be easily drawn from the two original studies. For example, the higher toxicity associated with the PD1-CTLA4 combination is well known to the melanoma community and, for this reason, PD1 combination is currently the standard of care.

Furthermore, the positive findings with the PD1-LAG3 inhibition have been described in detailed in the original paper in the New Engl J Med.

So, in conclusion, although I really appreciate the clarity and the methodology of this study, I feel that it is not adding new, practice-supporting information for the clinicians.

The authors themselves, state that there won't be any randomised trial between PD1-LAG3 vs PD1-CTLA4. In other words, the choice will generally be between PD1 inhibition alone vs PD1-LAG3.      

Author Response

  1. The authors of this paper analysed two randomised trials with dual checkpoint inhibition in patients with metastatic melanoma to compare the efficacy of PD1-LAG3 vs PD1-CTLA4.

The comparison is based on the extraction of IPD from KM curves of the original studies, through a dedicated algorithm. The study is presented clearly, the analysis is precise, and the discussion is comprehensive.

Reply: Thanks very much for your comment.

My main concern relates to the fact that the same conclusions can be easily drawn from the two original studies. For example, the higher toxicity associated with the PD1-CTLA4 combination is well known to the melanoma community and, for this reason, PD1 combination is currently the standard of care.

Furthermore, the positive findings with the PD1-LAG3 inhibition have been described in detailed in the original paper in the New Engl J Med.

So, in conclusion, although I really appreciate the clarity and the methodology of this study, I feel that it is not adding new, practice-supporting information for the clinicians.

Reply: Thanks very much for your comment. Yes, PD1 combination is currently the standard of care. However, which PD1 combination is the optimal for advanced melanoma is still under exploration. PD1-CTLA4 or PD1-LAG3? Our study, to some extent, indicated that PD1-LAG3 may be a better choice for advanced melanoma.

  1. The authors themselves, state that there won't be any randomised trial between PD1-LAG3 vs PD1-CTLA4. In other words, the choice will generally be between PD1 inhibition alone vs PD1-LAG3.

Reply: Thanks very much for your comment. Maybe, we didnot express well. Since nivolumab, ipilimumab and relatlimab were from the same company (Bristol Myers Squibb), and both CM67 and Rel47 acheived the primary endpoints. In the perspective of commercial interest, they would not fund a head-to-head randomized clinical trial to compare the two combinations. We had revised our statement (Page 12, Line 299).

Round 2

Reviewer 2 Report

Thanks for the clarifications provided.

I can confirm the overall good quality of the manuscript, although the research question and the results has low impact on clinical practice.

I would suggest to lessen the emphasis in the discussion/conclusions (e.g. in the sentence (Therefore, the present study is to some extent critical to help guiding clinical practice in selecting the immune checkpoint agents for advanced melanoma).

Regardless of this, the methodology is worth further application in other settings. 

Author Response

Thanks for your suggestion. We had lessen the emphasis in the discussion/conclusions as you suggest (Page 12, Line 301-302). 
